# Optimization of Electrolysis Parameters for Green Sanitation Chemicals Production Using Response Surface Methodology

**Nurul Izzah Khalid** [1], **Nurul Shaqirah Sulaiman** [1], **Norashikin Ab Aziz** [1,2,*],
**Farah Saleena Taip** [1], **Shafreeza Sobri** [3] and **Nor-Khaizura Mahmud Ab Rashid** [4]

1   Department of Process and Food Engineering, Faculty of Engineering, Universiti Putra Malaysia, UPM Serdang 43400, Selangor, Malaysia; nurul.izzah.khalid@gmail.com (N.I.K.); shaqirahiera@yahoo.com (N.S.S.); farahsaleena@upm.edu.my (F.S.T.)
2   Halal Products Research Institute, University Putra Malaysia, UPM Serdang 43300, Selangor, Malaysia
3   Department of Chemical and Environmental Engineering, Faculty of Engineering, Universiti Putra Malaysia, UPM Serdang 43400, Selangor, Malaysia; shafreeza@upm.edu.my
4   Department of Food Science, Faculty of Food Science and Technology, Universiti Putra Malaysia, UPM Serdang 43400, Selangor, Malaysia; norkhaizura@upm.edu.my
*   Correspondence: norashikin@upm.edu.my; Tel.: +603-9769-4302

**Abstract:** Electrolyzed water (EW) shows great potential as a green and economical sanitation solution for the food industry. However, only limited studies have investigated the optimum electrolysis parameters and the bactericidal effect of acidic electrolyzed water (AcEW) and alkaline electrolyzed water (AlEW). Here, the Box–Behnken experimental design was used to identify the optimum parameters. The tests were conducted with different types of electrodes, electrical voltages, electrolysis times, and NaCl concentrations. There were no obvious differences observed in the physico-chemical properties of EW when different electrodes were used. However, stainless steel was chosen as it meets most of the selection criteria. The best-optimized conditions for AcEW were at 11.39 V, 0.65 wt.% NaCl, and 7.23 min, while the best-optimized conditions for AlEW were at 10.32 V, 0.6 wt.% NaCl, and 7.49 min. The performance of the optimum EW (AcEW and AlEW) compared with commercial cleaning detergents for the food industry was then evaluated. The bactericidal activity of AcEW and AlEW was examined against *Escherichia coli* ATCC 10536 at different temperatures (30 °C and 50 °C) for 30 s. The results show that both AcEW and AlEW have the ability to reduce the *Escherichia coli* to non-detectable levels (less than 2 log CFU/mL).

**Keywords:** sustainable detergent; cleaning chemical; disinfection; sanitation; *Escherichia coli*; electrolyzed oxidizing water; electrolyzed reduction water; food soils; fouling deposit; response surface methodology

## 1. Introduction

Sanitation processes (i.e., cleaning and disinfection) are mandatory in the food industry to ensure that the production of safe food is always maintained. Due to this, daily sanitation is a common procedure in food production areas. However, this practice has several drawbacks such as high cost, high water usage, hazardous chemical effluents, and chemical residues in processing equipment which may affect food safety and quality. Hence, nowadays food producers and consumers demand green sanitation chemicals. Green sanitation is, in essence, using cleaning detergents and disinfection chemicals that are eco-friendly and do not emit pollutants.

Green cleaning detergents may seem weaker or less potent compared with available commercial cleaning chemicals in the market. However, green sanitizers such as electrolyzed water (EW) have shown significant potential in cleaning and sanitation for different types of food processing surfaces such as stainless steel [1–7], bamboo and wood [8], rubbers [9], and tiles [8,10]. Moreover, EW has several advantages that are favorable for a food manufacturer, especially small and medium companies [11]. Their budget allocation for sanitation and wastewater treatment are often at a minimum due to financial barriers. EW is reported to be inexpensive, uses on-site generation, and does not require a considerable amount of detergent storage space [1,4,12]. Moreover, in contact with organic matter or when diluted by tap water, osmosis water, and distilled water, acidic electrolyzed water (AcEW) will revert to its original form [1,4,13,14]. Thus, wastewater treatment costs can be reduced.

There are three main types of chemicals used for the cleaning and disinfection process: alkaline (e.g., sodium hydroxide), acidic (e.g., nitric acid), and disinfectants (e.g., hydrogen peroxide, sodium hypochlorite) [15]. Carbohydrate-based fouling deposits such as pink guava puree fouling deposit [16–18] and chili sauce fouling deposit [19] are alkaline soluble. In contrast, mineral-based fouling deposits such as calcium are developed mostly from beer [20,21] and dairy [22–24] and are acid soluble [17]. Various types of fouling deposits or food soils have different characteristics which require different cleaning processes and cleaning chemicals [15,16]. Alkaline electrolyzed water (AlEW) and AcEW are potential clean-label alternatives to the food industry for alkaline wash and acidic wash, respectively. Moreover, AcEW can also be an alternative disinfectant due to its antimicrobial properties for foodborne pathogens such as *Escherichia coli* [25–27], *Salmonella* [28–30] and *Listeria monocytogenes* [10,25,26].

Liquids at oxidation-reduction potential (ORP) ranges of 200 to 800 mV and −700 to 200 mV, respectively, are conditions in which aerobic bacteria and anaerobic bacteria can grow optimally [31]. Huang et al. [31] reported that at pH of 4 to 9, bacteria can also grow optimally. In this research, we target AcEW with a low pH range of 2.3 to 2.7, high ORP of more than 800 mV, and high chlorine content. The target for AlEW is set at a high pH of 11 to 12 and low ORP at below −700 mV. Hypochlorous acid (HOCl) [32], hypochlorite ion [32–34], and chlorine gas [32–34] are chlorine species in AcEW, and are responsible for the antimicrobial properties against different foodborne pathogens. At low pH of 2.3 to 2.7, AcEW sensitizes the bacterial cell's outer membrane and allows HOCl to penetrate the cells of the bacteria [35].

Several works in the literature have reported the effect of different electrolyzing parameters, which are the electrode types [36–39], electrode electrical conductivity [12,40], electrode's exchange current density [12,41], NaCl concentration [42,43], salt type [14], flow rate [42,43], temperature [42,43], electrical potentials [36,37], and electrolysis time [37] on the physical and chemical properties of AcEW. In contrast, studies on the usage of AlEW are still not well defined in the literature.

Hsu et al. [36] reported that a platinum–platinum (cathode–anode) pairing with higher electrical conductivity ($\sigma = 9.43 \times 10^6$ S/m) produced 3325 mg $Cl_2$/L, compared to a titanium–titanium pairing ($\sigma = 2.38 \times 10^6$ S/m), which only produced 3 mg $Cl_2$/L. Hsu et al. [36] concluded that the electrical conductivity of an electrode is mainly responsible for the production of chlorine. However, studies conducted by Khalid et al. [12] revealed that a titanium–silver pairing with higher electrical conductivity ($\sigma$ of silver = $6.3 \times 10^7$ S/m) produced a lower chlorine content compared with a titanium–stainless steel pairing ($\sigma$ of stainless steel= $1.45 \times 10^6$ S/m) (0.17 and 0.5 mg $Cl_2$/L, respectively). The quantity of chlorine generated is significantly low in the study conducted by Khalid et al. [12] as the lab-scale electrolyzing unit has a low current efficiency. According to Natarajan [41], higher exchange current density on electrodes will increase the hydrogen gas production. The exchange current density $i_o$ for $H^+$/$H_2$ reaction on platinum is about $10^{-2}$ A/cm$^2$, while the $i_o$ for $H^+$/$H_2$ reaction on zinc is about $10^{-11}$ A/cm$^2$. As such, reducing the hydrogen ions from the acidic electrolyte on the platinum electrode is easier, compared to the zinc electrode. The zinc electrode possesses a high hydrogen-overpotential (activation polarization). Lower $i_o$ leads to higher overpotential, while higher $i_o$ denotes lower overpotential. Thus, the reaction on zinc with higher $i_o$, are tends towards reversibility.

In our work, we are focusing on chlorine generation, and the work carried out by Natarajan [41] can be used as a reference.

Salt concentration, temperature, and water flow rate do not affect the electrical potential and power consumption of an electrolysis generator [42]. In the work by Hsu [43], salt concentration, temperature, and water flow rate have no impact on the pH and dissolved oxygen (DO) of AcEW. An increase in flow rate, however, reduces the efficiency of electrolysis (production of chlorine) [42,43]. The increase in the ORP and electrical conductivity of AcEW can be explained by the increase in the salt concentration [43]. The temperature has a minor effect on the total chlorine concentration [43]. Increasing the electrical potential will increase the electrical current flow and eventually increase the chlorine production [40]. Increasing the electrolysis time would then also increase chlorine production [37]. However, as the total chlorine production approaches the maximum level, the electrolysis time has no significant effect on the total chlorine production.

Various electrolytic variables have been reported to affect the properties of EW. To obtain the desired EW properties, many electrolysis parameters and their interaction must be considered simultaneously. Response surface methodology (RSM) was employed to understand the functional relationship between EW properties and the electrolysis parameters. RSM has been commonly used for optimizing and improving several processes. For processes using multiple variables, RSM can be employed to determine the interactions among the tested variables at different ranges. The RSM models generated are used to describe the effects of different variables on the response [37,44–48]. In this research, the process optimization of electrolyzing parameters for electrolyzed water (alkaline and acidic) was investigated using RSM.

The aim of this research is to identify the optimum electrolysis parameters (the types of electrodes, electrical voltages, electrolysis times, and NaCl concentrations) based on the physico-chemical properties (free chlorine, total chlorine, pH, dissolved oxygen, oxidation-reduction potential (ORP), electrical conductivity, and pH) of electrolyzed water (acidic and alkaline). The optimum condition was used to prepare the best EW. The bactericidal activity of the EW was evaluated against *Escherichia coli* at different temperatures. The results from this paper can be used as a guideline to find the best operational parameters of an EW generator and to formulate a suitable green cleaning solution, as different types of fouling deposits require different properties of cleaning solutions.

## 2. Materials and Methods

### 2.1. Laboratory-Scale Batch Electrolysis Unit

In this work, an electrolysis unit was used (Figure 1). This lab benchtop batch electrolysis unit was designed and installed at the Process and Food Engineering Department, Engineering Faculty, Universiti Putra Malaysia (UPM). This unit consists of an acrylic glass electrolysis chamber (which can be filled with up to 3 L of electrolyte for each chamber) and direct current (DC) power supply (PSW 30–36, with output voltage range of 0–30 V and output current of 0–36 A, GW Instek, Taiwan). The electrolysis unit consists of a cathode chamber and an anode chamber, which are separated by a membrane (polyester ultrafiltration membrane) that allows ion exchange. The DC power supply allows for the manipulation of the current inlet. Electrode slots were designed to ensure the electrode plates faced each other, which maximizes the ion exchange between electrodes. The gap between the electrodes is 15 mm. This slot allows the electrodes to stay in an immobile position and maintain the same gap throughout the electrolysis process.

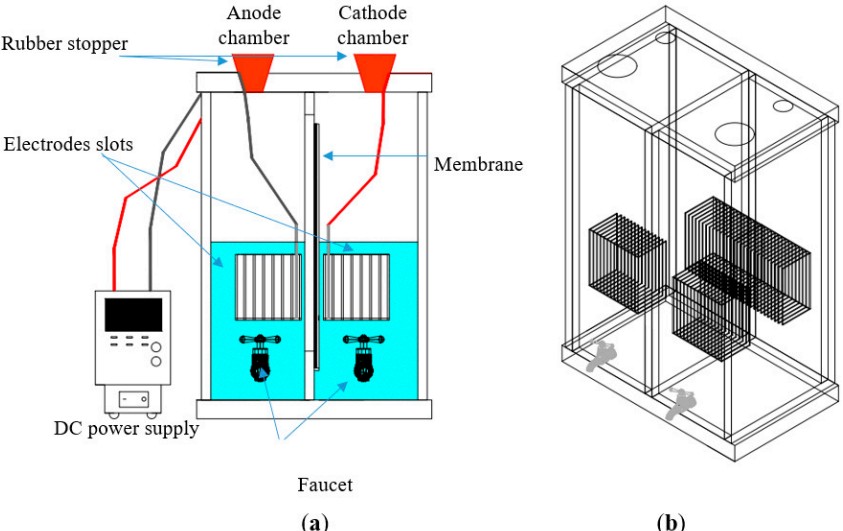

**Figure 1.** Schematic diagram of the laboratory-scale batch electrolysis unit: (**a**) front view and (**b**) side view.

## 2.2. Electrode Preparation

In this work, titanium, zinc, copper, and stainless steel 316 were used. The electrode materials were chosen based on several criteria: high electrical conductivity, corrosion resistivity, erosion resistivity, ability to catalyze the electrode reactions, affordability, and availability in the Malaysian market. The electrode is 100 mm wide, 100 mm long, and 1 mm thick. The electrodes were soaked in acetone 1% (EAM, Malaysia) for 3 h. The electrodes were then rinsed with distilled water. Next, the electrodes were soaked in ethanol 1% (R&M Chemicals, United Kingdom) for 10 min. After that, the electrodes were rinsed three times with distilled water and air dried for 24 h. The electrodes were placed in a desiccator prior to the electrolysis experiments. Titanium, zinc, copper, and stainless steel were used alternately as an anode. Stainless steel was placed at the cathode. All of these materials were supplied by Jetro Engineering Works, Malaysia.

## 2.3. Electrolysis Experiments

Sodium chloride (R&M Chemicals, London, United Kingdom) was diluted with distilled water at different dilution concentrations (0.05, 0.53, and 1 wt.% NaCl). The diluted salt solution was then poured into the electrolysis chambers (1.7 L evenly for each chamber). The cathode and anode electrodes were assembled and inserted inside the electrolysis chambers. Stainless steel was used as the cathode, and copper was used as the anode. Different inlet voltages (5, 10, and 15 V) were applied to the electrodes. Stoppers were used to ensure that the wires connecting the electrodes and DC power supply stayed static during the electrolysis process. The slots were also designed to ensure that the anode and cathode electrodes stayed facing each other and did not move. The electrolysis process was conducted at different electrolyzing durations (5, 7.5, and 10 min). At the end of the electrolysis process, AcEW and AlEW were collected from the anode and cathode chambers, respectively. The final current inlet was recorded as well. A similar procedure was repeated using zinc, titanium, and stainless steel as the anode.

## 2.4. Analytical Measurement of Electrolyzed Water

A compact photometer (PF-3, Macherey-Nagel, Düren, Germany) was used to measure the free chlorine. Powder pillow free chlorine (Visocolor, Macherey-Nagel, Düren, Germany), which contains *N*,*N*-diethyl-1,4-phenylene diamine (DPD) was added to a 5 mL sample (electrolyzed water). The available free chlorine reacts with DPD to form a red-violet dye, which can be analyzed

photometrically. The pH was measured through a portable pH meter (Thermo Fisher Scientific, Waltham, MA, USA). The ORP was measured using a handheld ORP meter (Boeco, Hamburg, Germany). The DO was measured using a handheld DO meter (Hanna Instruments, Woonsocket, RI, USA). The electrical conductivity (unit in mS) was measured with a portable conductivity meter (Aqualytic, Dortmund, Germany). A rapid test using an $H_2O_2$ test stick (Macherey-Nagel, Düren, Germany) was used to test the $H_2O_2$ content in AcEW. In this work, the chemical properties of sodium hypochlorite, NaOCl (R&M Chemicals, Southampton, UK), hydrogen peroxide, $H_2O_2$ (R&M Chemicals, Southampton, UK), and household bleach (Kuat Harimau, Melaka, Malaysia) were measured and compared with the optimized results of AcEW obtained from this research.

## 2.5. Electrolysis Efficiency for Chlorine Production

Current efficiency is, according to Faraday's law, the ratio of the actual mass of a substance liberated from an electrolyte by the current's passage to the theoretical mass liberated [12]. Equations (1) to (3) were used to calculate the theoretical total chlorine [12]. The mass of chlorine produced from the electrolysis experiments was calculated using Equation (4). Equations (5) to (7) were used to calculate current density, current efficiency, and electrical efficiency, respectively [36,37,40,49].

$$\textit{Charge transferred}, \; Q \; (A/s) = \textit{Electric current}, \; I \; (A) \times \textit{Electrolysis time}, \; t \; (s) \tag{1}$$

$$\textit{Number of mol electron transferred}, \; n \; (mol) = \frac{\textit{Charge transferred}, \; Q(A/s)}{\textit{Faraday constant}, \; F \; (C/mol)} \tag{2}$$

$$\textit{Theoretical mass of chlorine produced}, m \; (g) = n \; (mol) \times \textit{Molar mass} \left(\frac{g}{mol}\right) \tag{3}$$

$$\begin{aligned} \textit{Experimental mass of chlorine produced}, m \; (g) \\ = \textit{Experimental chlorine produced} \left(\frac{mg}{L}\right) \\ \times \textit{volume of water used for electrolysis} \; (L) \end{aligned} \tag{4}$$

$$\textit{Current density}, J \left(A/m^2\right) = \frac{\textit{Electrical current} \; (A)}{\textit{Effective surface area of anode}, A, \; (m^2)} \tag{5}$$

$$\textit{Current efficiency} \; (\%) = \frac{\textit{Experimental production of total chlorine} \; (mg)}{\textit{Theoretical production of total chlorine}, \; (mg)} \times 100\% \tag{6}$$

$$\textit{Electrical or Energy efficiency} \; (mg/kJ) = \frac{\textit{Total chlorine produced} \; (mg)}{\textit{Electrical energy consumed} \; (kJ)} \tag{7}$$

## 2.6. Experimental Design, Statistical Analysis, and Optimization

In this work, an experimental design called Box–Behnken (BBD) for response surface method (RSM) was applied using the Design Expert software (Stat-Ease Inc., Minneapolis, MN, USA). Wang et al. [46] and Cui et al. [48] mentioned that BBD is sufficient to fit a quadratic model which contains squared terms and products of two factors. Therefore, BBD is adequate to test the statistical validity of the fitted model and the model's lack of fit. For this three-factorial BBD, a total of 17 experiments were performed (12 experiments (each factor on three levels) plus five central experiments). In this particular study, the factors were voltages (5, 10, and 15 V), electrolyzing times (5, 7.5, and 10 min), and NaCl concentrations (0.05, 0.53, and 1 wt.%) as shown in Table 1. The predicted responses were pH, ORP, and free chlorine electrolyzed. Table 2 shows the desired target characteristics of EW.

**Table 1.** Box–Behnken design experiment factors and levels.

| Code | Factor | Level | | |
|------|--------|-------|-----|-----|
| | | −1 | 0 | 1 |
| A | NaCl Concentration (wt.%) | 0.05 | 0.53 | 1 |
| B | Electrolysis time (min) | 5 | 7.5 | 10 |
| C | Voltage (V) | 5 | 10 | 15 |

**Table 2.** Electrolyzed water (EW) desired target characteristics.

| Response | Types of Electrolyzed Water | | | |
|----------|--------|--------|--------|--------|
| | Acidic | | Alkaline | |
| pH | Minimum | 2–3 | Maximum | 11–12 |
| ORP | Maximum | >1000 mV | Minimum | <−800 mV |
| Free chlorine content | Maximum | >1 mg/L $Cl_2$ | | |

The following quadratic equation in Equation (8) was applied to fit the experimental data [47,48]:

$$Predicted\ response,\ R = b_0 + \sum_{i=1}^{n} b_i x_i + \left( \sum_{i=1}^{n} b_{ii} x_i \right)^2 + \sum_{i=1}^{n} \sum_{j=i+1}^{n} b_{ij} x_i x_j \tag{8}$$

where $b_i$ is the linear coefficient, $b_{ii}$ is the interaction coefficient, and $b_{ij}$ is the quadratic coefficient. $x_i$ and $x_j$ are the coded values of the electrolyzed water's variables (voltages, electrolyzing times, and NaCl concentrations), while the predicted responses are pH, ORP, and chlorine content. At a confidence level of 95% ($p < 0.05$), the correlations were considered statistically significant [47]. The coefficient of determination ($R^2$) and the adjusted determination coefficient ($R^2_{adj}$) were used to evaluate the adequacy and the reliability of the model [44,47].

*2.7. Experimental Validation*

The adequacy of the RSM models was validated. Two additional experiments were carried out by using the optimized condition of the best electrodes. Equation (9) was used to calculate the validity of this study.

$$Experimental\ validation = \frac{(Experimental - predicted)}{predicted} \times 100\% \tag{9}$$

*2.8. Bacterial Culture Preparation*

The optimized EW was used for microbiological testing. *Escherichia coli* ATCC 10536 obtained from the Food Microbiology Research Laboratory 1, Food Science and Technology Faculty, UPM was used as the test bacteria. Under refrigeration (at or below 4 °C), stock cultures were kept on nutrient agar (Oxoid, England, UK). Prior to the microbiological testing, the inocula were prepared from overnight culture growth on nutrient agar at 37 °C. The turbidity of the bacterial suspension was adjusted to a 0.5 McFarland standard (comparable to a bacterial suspension of approximately $10^8$ CFU/mL) [50]. A loopful of the culture was diluted in phosphate-buffer saline 0.1% (Sigma-Aldrich, St. Louis, MO, USA). The final concentration of $10^8$ CFU/mL was adjusted using a spectrophotometer according to the 0.5 McFarland standard.

*2.9. EW Treatment and Microbiology Analysis*

The antimicrobial activity of EW (AcEW and AlEW) against *Escherichia coli* ATCC 25922 was initially determined by mixing 2.0 mL of *Escherichia coli* ($1.5 \times 10^8$ CFU/mL) with 18 mL of EW in sterile universal bottles. Survival of *Escherichia coli* was determined immediately after mixing for 30 s,

by the spread-plate method with usage of Eosin Methylene Blue agar (Oxoid, England, UK) with serial dilution in sterile peptone water 0.1% (Oxoid, England, UK). Then, the plates were incubated for 24 h, and the colonies formed on the plates were subsequently counted. The result is expressed in log CFU/mL. Treatment using sterile distilled water was used as control. All treatments were conducted in triplicate.

### 2.10. Statistical Analysis

Using Minitab version 17.0, the selection of the best electrode was analyzed through two-way ANOVA. The types of electrodes and voltages were set as factors and the experimental chlorine, theoretical chlorine, current efficiency, and current were set as responses.

Meanwhile, the results of the microbiology tests were subjected to one-way ANOVA. The treatment was set as a factor and the surviving population of *Escherichia coli* was set as a response. The results are described as deviations of the means±standard error. ANOVA has been used to compare the means. Using the Tukey test, a significant difference between treatments was established at a significance level of *p* less than 0.05.

## 3. Results and Discussion

### 3.1. Selection of Anode and Cathode Materials

Table 3 shows the current flow, experimental total chlorine, theoretical total chlorine, current efficiency, and current density of different types of electrodes at 1 wt.% NaCl and 7.5 min electrolysis time. The efficiency of the electrode materials depends on the electrical conductivity, σ [12,40]. An excellent electrical conductor will be able to conduct a higher electrical current flow and increase the ion exchange during the electrolysis process. An increase in chlorine concentration would eventually increase the current efficiency [40]. The electrical conductivity of electrodes used in this work is shown in Table 4. Copper is the best electrical conductor compared with zinc, titanium, and stainless steel ($5.95 \times 10^7$, $1.82 \times 10^7$, $2.39 \times 10^6$, and $1.32 \times 10^6$ S/m, respectively). When electrolysis is carried out at 5 V, 1 wt.% NaCl, and 7.5 min, titanium shows the highest current efficiency and total experimental chlorine. Thus, the current efficiency of titanium at 5 V is the optimal option. However, at 15 V, 1 wt.% NaCl, and 7.5 min, copper shows the highest experimental chlorine production and current efficiency. Different electrodes with different electrical conductivity do not have a significant effect on the experimental and theoretical total chlorine. Moreover, the current flow during electrolysis is almost the same for all the electrodes (Table 3). For instance, at 15 V, the current flow for copper, zinc, titanium, and stainless steel are 4.09, 4.57, 4.35, and 4.53 A, respectively. Thus, the electrical conductivity of electrodes does not affect the chlorine production during electrolysis.

**Table 3.** Current efficiency for different electrodes at 1 wt.% NaCl and 7.5 min electrolysis time.

| Voltage (V) | Current (A) | Experimental Total Chlorine (mg/L) | Theoretical Total Chlorine (mg/L) | Current Efficiency (%) | Current Density, J (A/cm²) |
|---|---|---|---|---|---|
| | | **Copper** | | | |
| 5 | 0.82 | 0.66 ± 0.2 | 53.17 | 1.24 | 0.01 |
| 15 | 4.09 | 10.45 ± 2.33 | 265.21 | 3.94 | 0.05 |
| | | **Zinc** | | | |
| 5 | 0.64 | 0.46 ± 0.04 | 41.5 | 1.11 | 0.01 |
| 15 | 4.57 | 6.15 ± 1.77 | 296.33 | 2.087 | 0.06 |
| | | **Stainless steel** | | | |
| 5 | 1 | 0.35 ± 0.2 | 64.84 | 0.54 | 0.01 |
| 15 | 4.35 | 7.7 ± 0.14 | 282.07 | 2.73 | 0.05 |
| | | **Titanium** | | | |
| 5 | 0.68 | 1.8 ± 1.27 | 44.09 | 4.08 | 0.01 |
| 15 | 4.53 | 3.14 ± 0.66 | 293.74 | 1.07 | 0.06 |

**Table 4.** Electrical conductivity and electrical resistivity of different types of electrodes.

| Type of Electrode | Electrical Conductivity (S/m) | Electrical Resistivity ($\Omega$.m) | References |
|---|---|---|---|
| Copper | $5.95 \times 10^7$ | $1.68 \times 10^{-8}$ | [51] |
| Zinc | $1.82 \times 10^7$ | $5.48 \times 10^{-8}$ | [52] |
| Titanium | $2.39 \times 10^6$ | $4.18 \times 10^{-7}$ | [53] |
| Stainless steel 316 | $1.32 \times 10^6$ | $7.65 \times 10^{-7}$ | [54] |

The material's exchange current density, $i_o$ can also affect the production of chlorine [12,41]. A higher $i_o$ can produce a higher chlorine content [12]. The exchange current density of different electrodes used in this work is shown in Table 5. In Table 5, $i_o$ represents hydrogen production. We assume it to be similar to our work (chlorine production). A higher exchange current density on electrodes will increase the chlorine production. The exchange current density, $i_o$ for $2Cl^-/Cl_2$ reaction on copper is about $10^{-7}$ A/cm$^2$, while the $i_o$ for $2Cl^-/Cl_2$ reaction on zinc is about $10^{-11}$ A/cm$^2$. Chlorine ions from the electrolyte on the copper electrode can be reduced more easily compared with the zinc electrode. The zinc electrode possesses a high chlorine-over potential. The result shows that at 15 V, 1 wt.% NaCl, and 7.5 min electrolysis time, copper produces the highest amount of chlorine at 10.45 mg Cl$_2$/L with the highest current efficiency of 3.94%. The second-best option would be stainless steel, which produces 7.70 mg Cl$_2$/L with the highest current efficiency of 2.73%. This is followed by zinc (6.15 mg Cl$_2$/L with 2.08% current efficiency) and titanium (3.14 mg Cl$_2$/L with 1.07% current efficiency). Thus, the concentration of chlorine depends on the exchange current density.

**Table 5.** Exchange current density of different types of electrodes.

| Types of Electrodes | Exchange Current Density (A/cm$^2$) | References |
|---|---|---|
| Copper | $10^{-7}$ | [12,41] |
| Zinc | $10^{-11}$ | [12,41] |
| Titanium | - | - |
| Stainless steel | $10^{-6}$ (For Fe) | [12,41] |

In this work, the current efficiency is too low (between 0.54 and 4.08%) due to low experimental chlorine production. Low current efficiency might be due to the fouling (precipitate generated from corrosion) of a cathode, which hinders the electrolysis process and reduces its efficiency [40]. The corrosion reaction has become the domain limiting chlorine production. The fouling deposit generated and accumulated on the membrane's surfaces is limiting the ion exchange between the electrodes. Corrosion happens when metal (in this case, electrodes) and an electrolyte (NaCl solution) reacts during the electrolysis process. This corrosion is called electrochemical corrosion. The NaCl solution increases the conductivity of moisture around metal and accelerates the rusting process. Rust happens at the anode through a chemical process called oxidation in which metal atoms lose an electron, forming ions. The more efficient the electron flow from metal to oxygen, the quicker the metal rusts.

In this work, at higher electrical potential, the current flow increased and corrosion happened at the anode. During screening or preliminary work, at a higher electrical potential that was more than 15 V—for example, 20 V—the electrolyte at the anode chamber became brownish, proving the corrosion process. Thus, 15 V was used as the highest electrical potential. Even though copper produced the highest amount of chlorine, corrosion occurred severely at the anode chamber. The brownish color of the electrolyte might be due to some metal compounds (e.g., copper, zinc, and titanium) leaching out of the electrode into the electrolyte, which is not good for food industry application. No brownish solution was observed when the pairing of stainless steel–stainless steel was used. Stainless steel 316 is an iron-based alloy that contains 16.7% chromium, which enhances the stainless steel resistance to corrosion [55]. Thus, in this work, stainless steel 316 was chosen as the anode and cathode.

In electrolysis, the change of enthalpy ($\Delta H$) is the sum of internal energy change ($\Delta U$) and the work done by the system ($P\Delta V$) (Equation (10)). Work performed by the electrolysis system ($P\Delta V$) includes the generation of chlorine, hydrogen peroxide, oxygen, ozone, hydrogen, and many more. In this work, the low current efficiency might be due to the occurrence of other reactions at the same time during the electrolysis process, aside from the chlorine production. For instance, at the cathode, hydrogen is generated, while at the anode, hydrogen peroxide, oxygen, and ozone are generated [14]. In electrolysis, Gibbs free energy is known as the electrical energy input ($\Delta G$). Electrical energy input ($\Delta G$) is the difference between the enthalpy change ($\Delta H$) and entropy generated ($T\Delta S$) (Equation (11)). Electrical energy input ($\Delta G$) will decrease when the entropy is generated inside the electrolysis system. In this research, after the electrolysis process, the temperature of AcEW and AlEW increased slightly (0.1–2 °C). As heat is introduced to the electrolysis system, the $\Delta G$ and total energy input required ($\Delta H$) would then decrease. Thus, the current efficiency is reduced.

$$\Delta\ Enthalphy,\ \Delta H = \Delta U + P\Delta V \qquad (10)$$

$$\Delta\ Electrical\ energy\ input,\ \Delta G = \Delta H - T\Delta S \qquad (11)$$

All of the materials used for this study are widely available in most countries and are affordable. For instance, in Malaysia, zinc is the cheapest at RM 0.47/cm$^2$, followed by stainless steel, copper, and titanium (RM 0.85/cm$^2$, RM 0.95/cm$^2$, and RM 3.20/cm$^2$, respectively). Price was one of the core factors in selecting the electrode materials, as we were concerned about helping the SME food industries, which typically only have a minimum allocation for cleaning costs. Purchasing commercial food-grade cleaning detergents can be a burden for SMEs. The cost of cleaning chemicals can contribute up to 58% of total cleaning costs [17]. The selection of affordable and durable electrode materials will minimize the maintenance cost for the electrolyzing unit. By considering all the listed criteria, stainless steel is chosen as the best material for the cathode.

### 3.2. Influence of Electrical Potentials and NaCl Concentrations

Increasing electrical potential has increased the current flow (Table 3). A higher current flow increases the electron exchange. Some literature has suggested that the main driving force in the efficiency of the electrolysis process is the electrical potential [12,37,40]. The higher the potential of the cell, the higher the electrical current flowing through the electrolysis system and increasing the current density (Table 3). As the electrical potential increases from 5 V to 15 V, the current efficiency and current density subsequently increase as well, except for titanium. The theoretical chlorine production is low for titanium at 15 V. Thus, the current efficiency is decreased. In this work, the current efficiency only considers the generation of chlorine. During the electrolysis process and as the electrical potential was increased to 15 V, more bubbles were observed around the cathode and anode electrodes, indicating that there were other gases being generated. Thus, chlorine production was limited, and current efficiency reduced.

Tables 6 and 7 show the response surface design arrangement and response for AcEW and AlEW, respectively. Tables 8–10 show analyses of variance (ANOVA) for the developed response surface (RS) quadratic model for AcEW (pH, ORP, and chlorine content, respectively) obtained using the Design Expert software. The result shows that NaCl concentrations and voltages have significant effects ($p < 0.05$) on the pH, ORP, and chlorine content of AcEW. Tables 11 and 12 show the ANOVA for the developed RS quadratic model for AlEW (pH and ORP, respectively). Variables of NaCl concentrations and voltages also have significant effects on the pH and ORP of AlEW ($p < 0.05$). Figure 2 shows the response surface plot of the physico-chemical properties of EW due to the combined effect of NaCl concentration and voltage on pH. Figure 2a,b show that increasing the NaCl concentration and the voltage significantly, would reduce the pH of AcEW and increase the pH of AlEW, respectively, while Figure 3a,b show that the ORP of AcEW increases and AlEW reduces significantly ($p < 0.05$) when high voltage and high NaCl concentration are applied. At a high electrical potential, more electric

current passes through the electrolysis system and more chlorine is generated. The trend of chlorine profile is shown in Figure 4.

**Table 6.** Box–Behnken response surface design arrangement and response for acidic electrolyzed water (AcEW).

| Run | Voltage | NaCl Concentration | Electrolysis Time | pH | ORP (mV) | Chlorine Concentration (mg/L) |
|-----|---------|-------------------|-------------------|------|----------|-------------------------------|
| 1 | −1 | 0 | 1 | 4.45 | 378 | 0.21 |
| 2 | 0 | 0 | 0 | 2.73 | 1086 | 3.37 |
| 3 | 0 | 0 | 0 | 2.9 | 1069 | 3.26 |
| 4 | 1 | 0 | 1 | 2.78 | 987 | 7.45 |
| 5 | 0 | 1 | 1 | 3.03 | 482 | 3.11 |
| 6 | 1 | 0 | −1 | 3.89 | 930 | 1.78 |
| 7 | 0 | 0 | 0 | 2.49 | 1053 | 4.08 |
| 8 | −1 | 0 | −1 | 4.78 | 678 | 0.1 |
| 9 | 1 | −1 | 0 | 3.62 | 409 | 0.98 |
| 10 | −1 | 1 | 0 | 3.79 | 428 | 0.31 |
| 11 | 0 | −1 | −1 | 5.09 | 456 | 0.08 |
| 12 | 1 | 1 | 0 | 2.46 | 1109 | 7.3 |
| 13 | 0 | 0 | 0 | 2.73 | 1165 | 3.93 |
| 14 | −1 | −1 | 0 | 4.67 | 416 | 0.12 |
| 15 | 0 | 0 | 0 | 2.55 | 1104 | 3.98 |
| 16 | 0 | 1 | −1 | 2.91 | 1090 | 1.09 |
| 17 | 0 | −1 | 1 | 4.33 | 366 | 0.16 |

**Table 7.** Box–Behnken response surface design arrangement and response for alkaline electrolyzed water (AlEW).

| Run | Voltage | NaCl Concentration | Electrolysis Time | pH | ORP (mV) |
|-----|---------|-------------------|-------------------|-------|----------|
| 1 | −1 | 0 | 1 | 10.3 | 70 |
| 2 | 0 | 0 | 0 | 12.01 | −889 |
| 3 | 0 | 0 | 0 | 11.94 | −832 |
| 4 | 1 | 0 | 1 | 11.64 | −854 |
| 5 | 0 | 1 | 1 | 11.32 | −826 |
| 6 | 1 | 0 | −1 | 10.92 | 24 |
| 7 | 0 | 0 | 0 | 12.01 | −836 |
| 8 | −1 | 0 | −1 | 6.61 | 269 |
| 9 | 1 | −1 | 0 | 9.99 | 23 |
| 10 | −1 | 1 | 0 | 10.57 | 25 |
| 11 | 0 | −1 | −1 | 5.65 | 392 |
| 12 | 1 | 1 | 0 | 11.56 | −850 |
| 13 | 0 | 0 | 0 | 11.93 | −825 |
| 14 | −1 | −1 | 0 | 5.6 | 340 |
| 15 | 0 | 0 | 0 | 11.53 | −803 |
| 16 | 0 | 1 | −1 | 11.53 | −669 |
| 17 | 0 | −1 | 1 | 9.45 | −345 |

**Table 8.** Analysis of variance for the developed response surface quadratic model for pH of AcEW.

| Source | Sum of Square | Degree of Freedom | Mean Square | *F*-Value | *p*-Value |
|--------|--------------|-------------------|-------------|-----------|-----------|
| Model | 12.56 | 9 | 1.4 | 21.73 | 0.0003 |
| A-Voltage | 3.05 | 1 | 3.05 | 47.49 | 0.0002 |
| B-NaCl Concentration | 3.81 | 1 | 3.81 | 59.29 | 0.0001 |
| C-Electrolysis Time | 0.5408 | 1 | 0.5408 | 8.42 | 0.0229 |
| Residual | 0.4497 | 7 | 0.0642 | | |
| Lack of fit | 0.3433 | 3 | 0.1144 | 4.3 | 0.0963 |
| Corrected total | 13.01 | 16 | | | |

**Table 9.** Analysis of variance for the developed response surface quadratic model for oxidation-reduction potential (ORP) of AcEW.

| Source | Sum of Square | Degree of Freedom | Mean Square | *F*-Value | *p*-Value |
|---|---|---|---|---|---|
| Model | $1.668 \times 10^6$ | 9 | $1.853 \times 10^5$ | 34.2 | <0.0001 |
| A-Voltage | $2.945 \times 10^5$ | 1 | $2.945 \times 10^5$ | 54.35 | 0.0002 |
| B-NaCl Concentration | $2.672 \times 10^5$ | 1 | $2.672 \times 10^5$ | 49.31 | 0.0002 |
| C-Electrolysis Time | $1.107 \times 10^5$ | 1 | $1.107 \times 10^5$ | 20.43 | 0.0027 |
| Residual | 37930.95 | 7 | 5418.71 | | |
| Lack of fit | 30429.75 | 3 | 10143.25 | 5.41 | 0.0683 |
| Corrected total | $1.706 \times 10^6$ | 16 | | | |

**Table 10.** Analysis of variance for the developed response surface quadratic model for chlorine content of AcEW.

| Source | Sum of Square | Degree of Freedom | Mean Square | *F*-Value | *p*-Value |
|---|---|---|---|---|---|
| Model | 90.29 | 9 | 10.03 | 21.78 | 0.0003 |
| A-Voltage | 35.15 | 1 | 35.15 | 76.33 | <0.0001 |
| B-NaCl Concentration | 13.70 | 1 | 13.7 | 29.75 | 0.001 |
| C-Electrolysis Time | 7.76 | 1 | 7.76 | 16.85 | 0.0045 |
| Residual | 3.22 | 7 | 0.4606 | | |
| Lack of fit | 2.65 | 3 | 0.8829 | 6.14 | 0.056 |
| Corrected total | 93.51 | 16 | | | |

The electrical conductivity of the NaCl solution increased when higher NaCl concentrations were used. For instance, at 0.05 wt.% NaCl, the solution's electrical conductivity ranged from 0.86 to 0.98 Sm and its electrical conductivity was in the range of 16.00–16.23 Sm at 1 wt.%. This indicates that an increasing amount of concentrated NaCl allows more ion exchange, which in turn, increases the production of chlorine and the ORP for AcEW and reduces the pH; while increasing the pH and reducing the ORP in the cathode chamber. After electrolysis, the electrical conductivity did not noticeably change. Different electrical potentials and NaCl concentrations also had no effect on the DO and temperature of EW (data not presented). This is similar to the study by Hsu [43], who stated that increasing the NaCl concentration has no significant effect on the DO of EW.

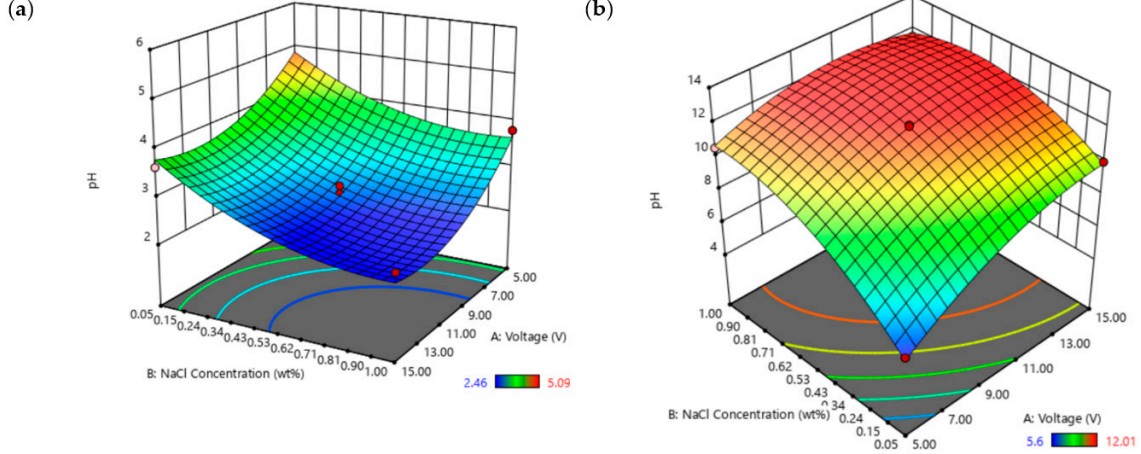

**Figure 2.** Response surface plot of the pH of: (**a**) AcEW and (**b**) AlEW due to the combined effect of the NaCl concentration and the voltage.

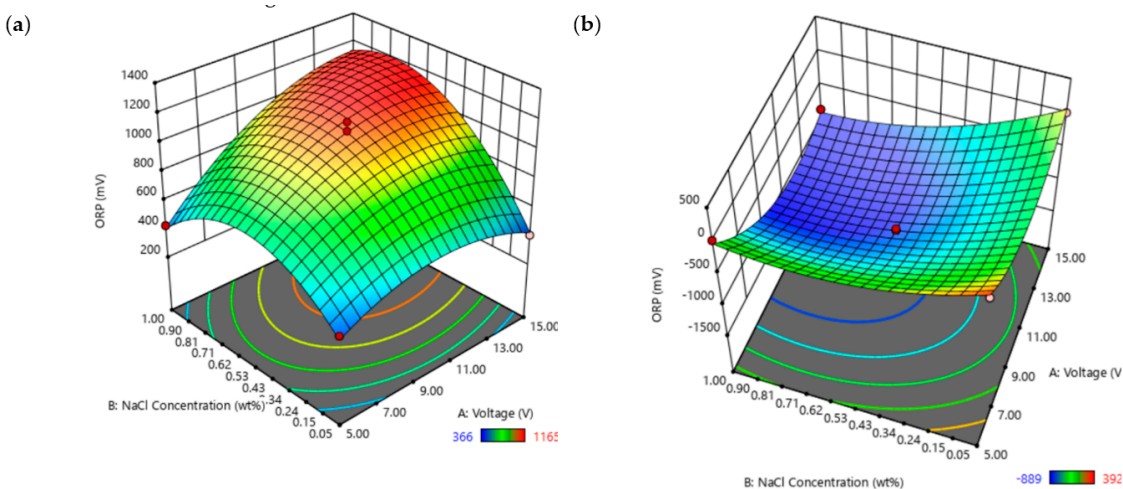

**Figure 3.** Response surface plot of the ORP of: (**a**) AcEW and (**b**) AlEW due to the combined effect of the NaCl concentration and the voltage.

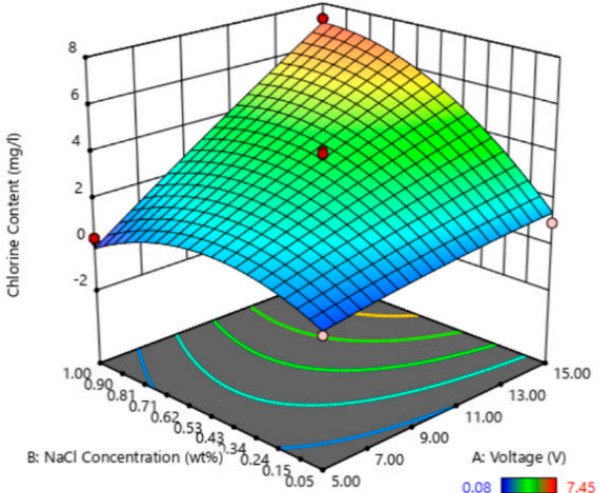

**Figure 4.** Response surface plot of the chlorine content of AcEW due to the combined effect of the NaCl concentration and the voltage.

*3.3. Influence of Electrolysis Time*

Studies on electrolysis time are essential as this information can be used to design the electrolysis system and to select suitable electrolyzing parameters [37]. Hsu et al. [37] stated that when chlorine generation reaches a saturation point, electrolysis time has no significant effect on the chlorine generation. In the work carried out by Hsu et al. [37], 10,498 mg $Cl_2$/L was produced at 7.5 V and 342 min electrolysis time. However, chlorine production was reduced as time continued to increase. This happens because the chlorine generation has reached its maximum level (saturation point) after being electrolyzed for a certain time interval. Khalid et al. [12] used three different electrolysis times of 10, 20, and 30 min. The study showed that at 9 V, 1 wt.% NaCl, and titanium–stainless steel pairing, an increase in the electrolysis time from 10 min to 20 min had a positive effect as 0.23 and 0.36 mg $Cl_2$/L were produced, respectively. Increasing the electrolysis time to 30 min did not increase the chlorine production as only 0.26 mg $Cl_2$/L was generated. Thus, it is crucial to determine the optimal choice of electrolysis time as extra electrolysis time can reduce chlorine production.

In this work, three different electrolysis times (5, 7.5 and 10 min) were investigated. Tables 8–10 show that the electrolysis time has a significant effect ($p < 0.05$) on the pH, ORP, and chlorine content of AcEW. The pH of AcEW also decreases when a longer electrolysis time is used (Figure 5a), while the

ORP (Figure 5b) and the chlorine production (Figure 5c) increase when a longer electrolysis time is used. A longer electrolysis time allows more time for current flow and ion exchange, which allows higher rate changes in the physicochemical properties (pH, ORP, and chlorine) of the EW. For instance, at 15 V, 0.53 wt.% NaCl, and 5 min electrolysis time (Run 6 in Table 6), the pH, ORP, and chlorine are 3.89, 930 mV, and 1.78 mg $Cl_2$/L, respectively. Increasing the electrolysis time to 10 min (Run 4 in Table 6), the pH, ORP, and chlorine change to 2.78, 987 mV, and 7.45 mg $Cl_2$/L, respectively. Theoretically, high chlorine content will lower the pH of EW [33,56]. As the chlorine increases from 1.78 mg $Cl_2$/L (5 min electrolysis time) to 7.45 mg $Cl_2$/L (10 min electrolysis time), the pH will eventually drop. The pH levels for the 5- and 10-min electrolysis times are 3.89 and 2.78, respectively. In this work, the current flow reading slowly increased from the initial readings. For instance, at 15 V, 0.53 wt.% NaCl, and 5 min electrolysis time (Run 6 in Table 6), the initial current flow starts at 1.93 ± 0.03 A and ends at 2.49 ± 0.02 A (0.56 A increment). When the electrolysis time is increased to 10 min, the initial current reading is 1.69 ± 0.05 A and ends at 2.31 ± 0.08 A (0.62 A increment). Thus, as the electrolysis time is increased, the current flow reading increases gradually. Different electrolysis times also have no effect on the DO, electrical conductivity and temperature of EW (data not presented). Tables 11 and 12 show that the electrolysis time has a significant effect ($p < 0.05$) on the pH and ORP of AlEW, respectively. The pH increases significantly (Figure 6a), while the ORP decreases significantly as well, as shown in (Figure 6b). For instance, at 10 V, 0.05 wt.% NaCl and 5 min of electrolysis time (Run 11 in Table 7), the pH and ORP are 5.65 and 393 mV, respectively. As the electrolysis time is increased to 10 min (Run 17 in Table 7), the pH and ORP are then 9.45 and −345 mV, respectively.

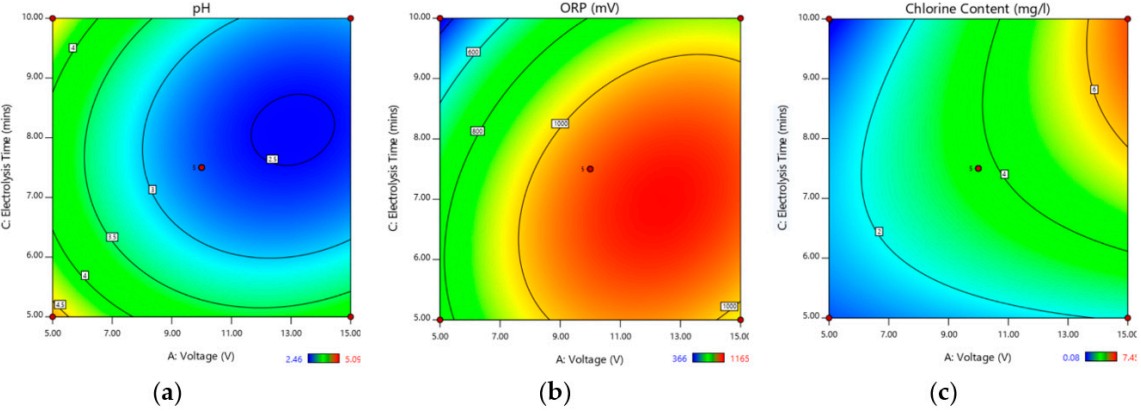

**Figure 5.** Response surface plot showing the effects of the electrolysis time and the voltage of AcEW on the (**a**) pH; (**b**) ORP and (**c**) chlorine content.

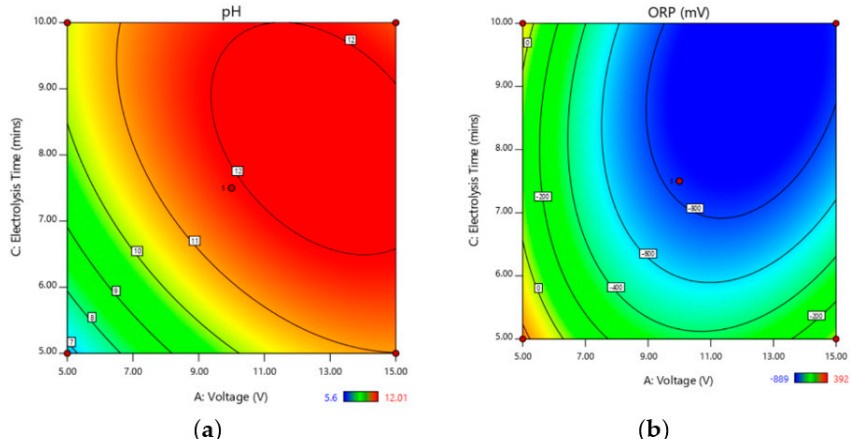

**Figure 6.** Response surface plot showing the effects of the electrolysis time and the voltage of AlEW on the (**a**) pH and (**b**) ORP.

### 3.4. Development of RS Model

ANOVA was used to test the developed model for predicting the chemical properties of AcEW, which are listed in Tables 8–10 (pH, ORP, and chlorine content, respectively). Tables 11 and 12 are the ANOVA for the developed RS quadratic model of the AlEW's pH and ORP. The Fisher *F*-value (Fisher's ratio value) test and lack of fit test was used to determine the significance and adequacy of the RS models. The significance of the RS models depends on the *F*-value and *p*-value [44]. The *F*-value must be bigger than the *p*-value to prove the significance of the results. In this work, the F-value for all the RS models was bigger than the *p*-value (less than 0.05). This indicates that all other models are much more significant. Lack of fit for all the RS models were not significant, indicating the adequacy of all the models [45]. This is shown in Tables 8–12.

**Table 11.** Analysis of variance for the developed response surface quadratic model for pH of AlEW.

| Source | Sum of Square | Degree of Freedom | Mean Square | *F*-Value | *p*-Value |
|---|---|---|---|---|---|
| Model | 76.79 | 9 | 8.53 | 136.21 | <0.0001 |
| A-Voltage | 15.21 | 1 | 15.21 | 242.77 | <0.0001 |
| B-NaCl Concentration | 25.53 | 1 | 25.53 | 407.48 | <0.0001 |
| C-Electrolysis Time | 8 | 1 | 8 | 127.71 | <0.0001 |
| Residual | 0.4385 | 7 | 0.0626 | | |
| Lack of fit | 0.2762 | 3 | 0.0921 | 2.27 | 0.2226 |
| Corrected total | 77.23 | 16 | | | |

**Table 12.** Analysis of variance for the developed response surface quadratic model for ORP of AlEW.

| Source | Sum of Square | Degree of Freedom | Mean Square | *F*-Value | *p*-Value |
|---|---|---|---|---|---|
| Model | $3.980 \times 10^6$ | 9 | $4.423 \times 10^5$ | 129.28 | <0.0001 |
| A-Voltage | $6.968 \times 10^5$ | 1 | $6.968 \times 10^5$ | 203.68 | <0.0001 |
| B-NaCl Concentration | $9.316 \times 10^5$ | 1 | $9.316 \times 10^5$ | 272.32 | <0.0001 |
| C-Electrolysis Time | $4.856 \times 10^5$ | 1 | $4.856 \times 10^5$ | 141.95 | <0.0001 |
| Residual | 23946.75 | 7 | 3420.96 | | |
| Lack of fit | 19916.75 | 3 | 6638.92 | 6.59 | 0.05 |
| Corrected total | $4.004 \times 10^6$ | 16 | | | |

The quadratic models can be used to predict the pH, ORP, and chlorine production of AcEW (Equations (12)–(14), respectively) while the quadratic models (Equations (15) and (16)) can predict the pH and ORP for AlEW. All the coefficients of determination ($R^2$) are more than 0.90, indicating that the model could not expect less than 10% of the total variation of the data obtained. The adjusted determination coefficient ($R^2_{adj}$) values for the models are more than 0.90, suggesting the model is reliable in predicting experimental results. Hence, all the models obtained through the usage of the RSM can be used to predict the chemical properties of EW.

For AcEW:

$$pH = 2.68 - 0.6175A - 0.69B - 0.26C - 0.07AB - 0.195AC + 0.22BC + 0.545A^2 + 0.41B^2 + 0.75C^2 \tag{12}$$

$$ORP = 1095.4 + 191.8A + 182.75B - 117.63C + 172AB + 89.25AC - 129.5BC - 180.08A^2 - 324.83B^2 - 172.07C^2 \tag{13}$$

$$Chlorine\ content = 3.72 + 2.1A + 1.31B + 0.985C + 1.53AB + 1.39AC + 0.485BC - 0.1357A^2 - 1.41B^2 - 1.2C^2 \tag{14}$$

For AlEW:

$$pH = 11.88 + 1.38A + 1.79B + C - 0.85AB - 0.7425AC - BC - 1.04A^2 - 1.42B^2 \\ - 0.9795C^2 \tag{15}$$

$$ORP = -837 - 295.13A - 341.25B - 246.38C - 139.5AB - 169.75AC + 145BC \\ + 480.38A^2 + 241.12B^2 + 233.88C^2 \tag{16}$$

### 3.5. Experimental Validation

Based on data associated with the numerical optimization and using a stainless steel–stainless steel electrode pairing, the best-optimized conditions for AcEW would be 11.39 V, 0.65 wt.% NaCl and 7.23 min, to produce pH of 2.43, 1184 mV and 4.5 mg $Cl_2$/L. Meanwhile, the best-optimized conditions for AlEW were 10.32 V, 0.6 wt.% NaCl and 7.49 min, to produce a pH of 12.19 and −900 mV. To validate this data, an experiment was carried out using the optimized conditions and was repeated twice.

Table 13 shows the experimental validation results for AcEW and AlEW, respectively. From Table 13, the average errors for pH, ORP, and chlorine content are found to be well below the predicted values at only 13%, 1%, and 6%, respectively, for AcEW. The error might have occurred due to the dirty membrane (corrosion product that accumulated on the membrane surface), the brown precipitate (corrosion product) deposited on the ion-exchange membrane surfaces, and the limitation of ion exchange. Thus, the generation of chlorine was reduced which reduced the rate of changes in chemical properties such as ORP and pH. The membrane was used several times during the electrolysis process. This might be the reason for the inconsistency in the EW properties. When the current flow became too low or the membrane considerably dirty, we changed the membrane. However, the membrane change was occasional, as it increased the cost.

The average percentage errors for pH and ORP are 7% and 9%, respectively, in relation to AlEW, as shown in Table 13. Based on the authors' reading of several works which used a commercial electrolyzing unit, the ORP value for AlEW can vary from −795 to −867 mV [1,4,27,29,57]. The lowest ORP for AlEW is −867 mV, which was reported by Xie et al. [58]. In our work, the values of ORP for AlEW obtained from actual data (−817 ± 8.49 mV) are similar to those generated using various commercial electrolyzing units. It can be concluded that the derived regression model established through these optimizing parameters is able to optimize the electrolyzing values for achieving pH, ORP, and chlorine accurately.

**Table 13.** Experimental validation of pH, ORP and free chlorine using stainless steel as cathode for AcEW and AlEW.

| Type of EW | | pH | Error (%) | ORP | Error (%) | $Cl_2$ (mg $Cl_2$/L) | Error (%) |
|---|---|---|---|---|---|---|---|
| AcEW | Predicted | 2.43 | 13 | 1184 | 1 | 4.47 | 6 |
| | Actual | 2.74 ± 0.03 | | 1168 ± 3.54 | | 4.21 ± 0.05 | |
| AlEW | Predicted | 12.19 | 7 | −900 | 9 | | |
| | Actual | 11.38 ± 0.01 | | −817 ± 8.49 | | | |

### 3.6. Comparison of AcEW with Available or Common Disinfectants

The physico-chemical properties of other disinfectants were measured and compared with AcEW obtained from this study (Table 14). The measurement was taken at room temperature. Hydrogen peroxide, $H_2O_2$ is able to act as an antimicrobial agent [34,38,39,58]. It can be used as a disinfectant for food-contact surfaces [59]. Sodium hypochlorite is also one of the common sanitizers used for cleaning and sanitation in the food industry due to its high chlorine content [4]. The label on a $H_2O_2$ bottle usually states that it contains 10% available chlorine. However, only 1.59 mg/L free chlorine ($Cl_2$) was detected. AcEW contains 7.2 mg/L $Cl_2$ and is comparable with $H_2O_2$. Moreover, AcEW also contains $H_2O_2$. A rapid test using an $H_2O_2$ test stick (Macherey-Nagel, Düren, Germany) was used to test the

$H_2O_2$ content in AcEW. The results show that AcEW contains 10 to 30 mg/L of $H_2O_2$. This shows that AcEW obtained from this study can be used in the food industry as a green disinfectant.

**Table 14.** Comparison of characteristics of other types of disinfectants with EW.

| Type of Disinfectant | pH | ORP (mV) | Chlorine Content (mg/L) | Dissolved Oxygen (mg/L) | Electrical Conductivity (mS) |
|---|---|---|---|---|---|
| AcEW * | 2.46 ± 0.07 | 1148 ± 16.97 | 8.25 ± 0.21 | 5.15 ± 0.07 | 9.16 ± 0.08 |
| AlEW * | 11.72 ± 0.01 | −832 ± 6.36 | - | 5.05 ± 0.07 | 9.33 ± 0.01 |
| Sodium hypochlorite | 12 | 418 | 1.59 | 4.8 | >20 |
| Hydrogen peroxide | 2.11 | 502 | 0.55 | 21 | 0.43 |
| Commercial bleach | 11.6 | 565 | 0.16 | 8.9 | >20 |

\* Obtained from this work.

### 3.7. Treatment of EW on Escherichia coli

The efficiencies of sanitizers (AcEW, AlEW, and sterile distilled water (control)) against *Escherichia coli* ATCC 10536 are shown in Table 15. AcEW (chlorine: 4.21 ± 0.05 mg/L, ORP: 1168 ± 3.54 mV, pH: 2.74 ± 0.03) and AlEW (ORP: −817 ± 8.49 mV, pH: 11.38 ± 0.01) exhibits a strong bactericidal activity against *Escherichia coli*. After 30 s of treatment using EW (AcEW and AlEW), no survival of *Escherichia coli* is detected (less than 2 log CFU/mL). Increasing the temperature to 50 °C does not have any significant effect ($p < 0.05$). Cleaning at an ambient temperature of 30 °C is sufficient to disinfect *Escherichia coli* ATCC 10536. Thus, it is recommended to use EW at 30 s and 30 °C which is sufficient to disinfect *Escherichia coli* ATCC 10536.

**Table 15.** Effect of different cleaning treatment in disinfecting *Escherichia coli* ATCC 10536.

| Treatment | Temperature (°C) | Surviving Populations ($\log_{10}$ CFU/mL) |
|---|---|---|
| Sterile distilled water | 30 | 6.67 ± 0.31 [a] |
| | 50 | 6.96 ± 0.11 [a] |
| Acidic electrolyzed water | 30 | ND [b] |
| | 50 | ND [b] |
| Alkaline electrolyzed water | 30 | ND [b] |
| | 50 | ND [b] |

Notes: ND means non-detectable level which is less than 2 log CFU/mL. Different superscript letters mean that the values are significantly different when tested with least significance differences test.

### 4. Conclusions

Through this study, the optimal conditions of electrolysis for producing AcEW and AlEW were obtained using RSM. The results show that the quadratic models with $R^2$ and $R^2_{adj}$ values of more than 0.90 adequately described and predicted the responses (pH, chlorine and ORP) under tested conditions. Electrical voltage, NaCl concentration, and electrolysis time have significant effects ($p < 0.05$) on the pH, ORP, and chlorine production of EW. For AcEW, according to the *p*-values, the effects of the variables on pH follow the order of NaCl concentration > voltage > electrolysis time; and the effects of the variables on chlorine and ORP follow the order of voltage > NaCl concentration > electrolysis time. According to the *p*-values, the effects of the variables on pH and ORP for AlEW are the same. The optimal conditions obtained for AcEW are at 11.39 V, 0.65 wt.% NaCl, and 7.23 min, while the optimal conditions for AlEW are at 10.32 V, 0.6 wt.% NaCl, and 7.49 min. The experimental data and the predicted values of response variables are extremely close with percentage errors of less than 13%. Therefore, the developed polynomial models are powerful, and the model can be used to predict target electrolysis performance. Stainless steel was chosen as the best electrode for both anode and cathode. The optimum electrolyzed parameters were used for antimicrobial testing. The results show that

both AcEW and AlEW are effective in inactivating *Escherichia coli* and could be used as a disinfectant agent alternative for reducing the bacterial contamination of food processing surfaces. More practical research should be carried out to evaluate AcEW and AlEW as a potential clean-label alternative in the food industry for cleaning detergents.

**Author Contributions:** Conceptualization, N.I.K. and N.A.A.; Methodology, N.I.K.; Data curation, N.I.K. and N.S.S.; Writing—original draft, N.I.K.; Supervision, N.A.A., F.S.T., S.S. and N.-K.M.A.R; Project administration, N.A.A.; Reviewing—final manuscript, N.A.A.; Reviewing, N.A.A., F.S.T., S.S. and N.-K.M.A.R. All authors have read and agreed to the publisher version of the manuscript.

**Funding:** This research was funded by Universiti Putra Malaysia, Malaysia through High Impact Putra Grant, grant number "9658400" and "The APC was funded by Universiti Putra Malaysia".

**Conflicts of Interest:** The authors have declared no conflict of interest.

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
