# Peer review of "Optimization of Electrolysis Parameters for Green Sanitation Chemicals Production Using Response Surface Methodology"

_processes, doi:10.3390/pr8070792_

Round 1

Reviewer 1 Report

The present manuscript nicely shows how statistical design of experiments can be used to optimise experimental parameters with limited experimental runs but maximum results. All experiments are well described and the conclusion are convincing. However, authors should be careful with significant digits. Minimum remarks are:

Page 4, electrolysis unit: What was the kind of membrane, please specify, because numerous membranes allowing ion exchange exist.

Equation 1: It must be „Electrolysis time“

Equation 8: xj ist missing.

Bacterial culture preparation: Where was the PBS from? What was the concentration of the buffer components?

Page 8, line 239: Delete „increased“.

Table 3: It is meaningful to give so much digits? What about the standard deviation of the values? Please give only as much digits as meaningful (the same holds true for all other Tables and Equations 12 to 16).

Symbols: All symbols should be given in italics (according to IUPAC recommendations).

Reviewer 2 Report

This work presents an optimization strategy for Electrolyzed water (EW). The work is of interest to sectors striving to implement green and economic sanitation at its best. While the organization of the manuscript and writing is good, I suggest the following points to be addressed before publications:

  1. Consider revising the title for better clarity
  2. Make a language check once more.
  3. “The aim of this research was to identify the optimum electrolyzed parameters (the type of electrodes, electrical voltage, electrolysis time, and NaCl concentrations) based on physico-chemical properties (free chlorine, total chlorine, pH, dissolved oxygen, oxidation-reduction potential (ORP)”. Physico-chemical properties of what? Please make a clear presentation when you write.
  4. What type of membranes are employed in the Electrolyzer unit? How many cell pairs?
  5. In the Box-Benkend design, the range of NaCl concentration parameters in small: what's your justification? How sensitive are the final water parameters to the initial NaCl concentration?
  6. Did you use Design experiment software? If so, why not mentioned in the methods part?
  7. 9.l.264. “The result shows that at 15 V, 1.0% 265 NaCl, and 7.5 minutes electrolysis time, copper can produce the highest amount of chlorine of 10.45 266 mg Cl2/L with the highest current efficiency of 3.94%. The second-best option would be stainless 267 steel, which produces 7.70 mg Cl2/L chlorine with the highest current efficiency of 2.73%.”……I don’t understand why the trend in current efficiency for the different electrodes is not coherent with the exchange current density?
  8. 9.l.284. “In this work, at higher electrical potential, the current flow increased, and corrosion happened at the anode”. What about gas crossovers at high potential?
  9. 10.l.316. “3.2 Influence of electrical potentials and NaCl concentrations on the EW’s physical and chemical properties”. Why such a long title? Reduce throughout the manuscript
  10. 15.l.352. “The pH of AcEW also decreased when longer electrolysis time was used (Figure 393 5a), while the ORP (Figure 5b) and the chlorine production (Figure 5c) increased when longer 394 electrolysis time was used.” Justify why the reason behind this trend? There is also a stability issue with time in some similar systems. How stable is the current/voltage in your system?
  11. 18.l.446. “From Table 13, the average errors for pH, ORP, and chlorine content were found to be well below the p. predicted value at only 13%, 1%, and 6%, respectively, for AcEW.”..How valid is the 13% difference in the experimental and predicted result?
